# Crystallization Kinetics of Modified Basalt Glass

**DOI:** 10.3390/ma13215043

**Published:** 2020-11-09

**Authors:** Yonglin Huo, Guilu Qin, Jichuan Huo, Xingquan Zhang, Baogang Guo, Kaijun Zhang, Jun Li, Ming Kang, Yanhui Zou

**Affiliations:** 1State Key Laboratory of Environment-friendly Energy Materials, Southwest University of Science and Technology, Mianyang 621010, China; hyl20201016@163.com; 2Institute of Chemical Materials, China Academy of Engineering Physics, Mianyang 621900, China; qgl13653595246@163.com (G.Q.); zkj7220170174@163.com (K.Z.); 3Analysis and Testing Center, Southwest University of Science and Technology, Mianyang 621010, China; baogangguo@swust.edu.cn; 4Sichuan Tianyi optical glass Co., Ltd., Sichuan Fiberglass, Deyang 618000, China; lijun19830501@aliyun.com (J.L.); kangming_652@163.com (M.K.); zouyanhui_ty@163.com (Y.Z.)

**Keywords:** basalt glass, crystallization activation energy, crystal growth index, crystallization kinetics, crystalline phase

## Abstract

As the raw material for the production of basalt continuous fibers in Sichuan, basalt glass (BG) and modified basalt glass (MBG) were prepared by the melt quenching method with the basalt and chemically modified basalt, respectively. The crystallization kinetics of BG and MBG were investigated by differential scanning calorimetry (DSC) according to Kissinger methods. The results revealed that it is difficult for both glasses to crystallize at a high temperature. In addition, the crystallization activation energy of MBG is much higher than that of BG, which indicates that MBG is more difficult to crystallize than BG. The crystalline phases seemed to be formed from the surface of the two glasses. The morphologies and crystal structure of the crystalline phases in the heat-treated BG and MBG were analyzed by scanning electron microscope (SEM/EDX) and XRD. It was found that only a small amount of crystalline phase can be observed in the MBG, which indicates that the crystallization ability of the MBG was greatly suppressed. Results of this initial investigation indicate that chemical modification can effectively suppress the crystallization tendency of basalt glass and improve its thermal stability, which opens up an effective way for the industrial scale and stable production of basalt fiber.

## 1. Introduction

The development of high-performance fibers has always been a hot spot in the research of fiber materials. Basalt fiber is a continuous fiber made by melting basalt above 1450 °C and drawing it at high speed by using platinum-rhodium alloy drawing bushing. The processing process is green and environment-friendly, and the processed fiber has excellent physical and chemical properties such as high tensile strength, high elastic modulus, high temperature resistance, erosion resistance, heat insulation, sound insulation, etc. It is widely used in fiber reinforced composite materials, aerospace, national defense and military industry, vehicle and ship manufacturing, petrochemical industry, civil engineering and transportation, marine engineering and other fields, and it has a huge market prospect [1,2,3].

At present, countries with the basalt fiber industrial production technology include Ukraine, Russia, Georgia, China, South Korea, Austria, Belgium, Germany, etc., but the development of basalt fiber industrial production technology all over the world is in the primary stage. The pure natural basalt melt has poor heat permeability. In the process of preparing the fiber, the crystallization upper limit temperature is high and it is easy to crystallize, which means that the material property is unstable. In addition, the chemical composition of natural basalt isnot fixed, so it is difficult to realize the large-scale production of basalt continuous fiber. The chemical composition of basalt should be modified for the industrial development of basalt fibers. The basalt glass is easy to crystallize during the production process. In order to produce basalt fibers with good performance on an industrially stable scale, the TFe content in basalt glass is reduced for modification [4,5,6] to suppress the crystallization behavior during the production of basalt fiber. Currently, researchers pay more attention to the high temperature crystallization temperature and crystalline phases such as spinel (MgAl_2_O_4_), peridot ((Mg,Fe)_2_SiO_4_), and iron and titanium oxide (FeTiO_x_). Walker et al. [7] conducted a study on the crystallization of basalt on the moon and found that the phases crystallize in the order olivine, chromium spinel, pyroxene, plagioclase, and ilmenite during the equilibrium crystallization. Namur O et al. [8] thought that the Sept Iles layered intrusion crystallized from a ferrobasaltic parent magma. The sequence of crystallization in the Sept Iles layered series is: Plagioclase and olivine, followed by magnetite and ilmenite, then Ca-rich pyroxene, and finally apatite. Tripoli B et al. [9] found that deformation enhanced the crystallization kinetics in basaltic magmas. They observed that the nucleation and growth rates of spinels and Fe-Ti oxides increase when deformation is applied to a basaltic melt. Gu et al. [10] found that with the increase of SiO_2_, the crystallization temperature range of basalt melts became smaller. Fan and Tong [11,12] found that the greater the mineral melt viscosity was, the better the mechanical and chemical properties of the fibers were. The crystallization kinetics of basalt glass and the effective inhibition of basalt glass crystallization through modification have not been reported in relevant literatures. The main mineral composition of basalt used for fiber production in Sichuan, China was analyzed by a micro-image analyzer and X-ray diffraction (XRD), and the crystallization kinetics of basalt glass and modified basalt glass were studied using differential scanning calorimetry (DSC). In addition, the morphology and chemical composition of unmodified and modified basalt glass after the crystallization heat treatment were studied by XRD, SEM, and EDS. These are of great guiding significance to the screening of basalt for continuous fiber production and the large-scale and stable industrial production of continuous basalt fiber [13,14,15,16,17]. 

## 2. Materials and Methods

### 2.1. Experimental Materials

The main factor determining the high temperature stability of basalt fibers is their crystallization behavior and the crystallization ability primarily depends on the fiber chemical composition. In this work, the modified basalt glasses were obtained by correction of the oxide composition of the natural basalt rocks by adding a proper amount of pyrophyllite (Al_2_(Si_4_O_10_)(OH)_2_), diopside (CaMg(SiO_3_)_2_), and calcium carbonate. The chemical composition of the raw materials was given in Table 1. 

For this work, a series of modified basalt glasses were synthesized by the melt quenching method. The main indicator is the modification of the iron content. Compositions of batches were given in Table 2.

The raw materials of basalt rocks, pyrophyllite, diopside, and calcium carbonate were mixed by ball-milling for 3 h. About 50 g of the mixture was heated in alumina crucibles to dissociate the carbonates and then melted isothermally for 3 h at 1450 °C to form homogeneous melts, followed by quenching of the melts into water. The quenched samples were crushed and melted again at 1450 °C for 2 h to ensure homogeneity, then poured onto a pre-heated steel plate. 

The crystallization kinetics and crystallization tendency of these modified basalt glasses were studied. The obtained results indicated that the content of iron has significantly impacted on the thermalstability of basalt glass. Taking into account the properties of basalt fiber and production cost, we developed a complete set of production technology of continuous basalt fiber by the tank furnace based on anoptimized glass chemical composition (MBG-2 in Table 3). The analysis of the continuous basalt fiber product produced by the world’s first 10,000 ton-class tank furnace revealed that the production cost can be reduced by more than 40% and the quality of continuous basalt fiber can be improvedby more than 30% based on the optimized glass chemical composition. Therefore, in this work, the crystallization kinetics and crystallization tendency studies were mainly performed on MBG-2. 

### 2.2. Material Characterization

The lithofacies microscopic analysis of 0.03 mm thick basalt slices were performed using the LEITZ LABORLUX 20pol image analyzer (Leitz, Wertheim, Germany) and Mias 2000 sensitive view analyzer (Institute of Computer Image and Graphics, Sichuan University, Sichuan, China). Crystalline phases were characterized by the X’ Pert Pro multifunctional x-ray diffractometer (PANalytical B.V., Heracles Almelo, The Netherlands).

Crystallization kinetics of the samples were determined by the differential temperature analysis (DTA, SDT Q600, TA Instruments Inc., New Castle, DE, USA). According to the crystallization peaks from the DSC curve of basalt glass, crystallization kinetics were obtainedbased on the relationship between the crystallization temperatures and the heating rates. The BG and MBG were heat-treated at their crystallization temperatures, respectively. The crystal structure of all the samples were examined by X-ray diffraction analysis (XRD, Bruker D8 ADVANCE (Karlsruhe, Germany) with Cu-Kα radiation, λ = 1.5418 Å). The scanning speed was 8°/min and the scanning range was 3°–80°. The chemical composition and surface morphology of the crystallized samples were determined by a Carl Zeiss AG Ultra 55 scanning electron microscope (SEM, Jena, Germany).

## 3. Results and Discussion

### 3.1. Petrographic Features of the Basalt Rocks

Basalt is a grey to black, fine-grained volcanic rock that is formed by magmatic movement and subsequent sudden cooling of magma in atmospheric conditions. The basaltic rocks can be separated into two main groups: Subalkaline and alkaline. The subalkaline group is constituted by tholeiitic and calcalkaline basalts, whereas the alkaline group is constituted by the alkaline and alkaline-olivine basalts. The tholeiite basalts are usually used forpetrurgy due to their higher viscosity. While, alkaline basalts are more suitable for the production of glass-ceramic materials due to their low viscosity. In this work, the basalt glass and modified basalt glass were prepared with basalt rocks as the main raw materials. In order to better understand the effect of the raw material on the properties of the obtained basalt glass, we investigated the petrographic features of the used basalt rocks. Basalt rocks obtained from the Central Sichuan region of China were used in this study. 

Figure 1 shows the XRD patterns of the basalt rocks. Plagioclase ((Ca,Na)((Si,Al)_4_O_8_), PDF#83-1939), augite (Ca(Mg,Fe,Al)((Si,Al)_2_O_6_), PDF#41-1483), quartz (SiO_2_, PDF#89-1961), chlorite ((Mg,Al)_6_((Si,Al)_4_O_10_(OH)_8_), PDF#22-0712), and magnetite (Fe_3_O_4_, PDF#75-0033) phases were determined. Figure 2 shows microscopic features of the basalt rocks. The rate of decomposition of the basalt rocks is about 70%. The transparent minerals are mainly plagioclase (Pl), pyroxene (Px), and a small amount of quartz (Qtz), chlorite (Chl), and olivine, whereas the metal minerals are mainly magnetite (Mag) with a small amount of pyrite (Py) and chalcopyrite. The size of monomer minerals is about 0.004–0.03 mm, and the size of connective is about 0.01–0.04 mm. Plagioclase (Pl) is tabular, lath, and granular, with a slight clay alteration on the surface, and iron mineral inclusions can be seen in a small amount of particles—Pyroxenes (Px) are mostly monomers in columnar, strip, granular, and irregular shapes. A small amount of particles is associated with iron minerals and other gangue minerals. Some particles can be seen in iron mineral inclusions with chloritization and serpentinization alteration. Magnetite (Mag), ilmenite (Py), pyrite, and chalcopyrite are fine-grained and irregular, mostly dissociated monomers, and a small amount of them are connected or wrapped in gangue minerals.

Figure 3 shows DSC curves of the BG and MBG measured at different heating rates. The BG has two obvious crystallization peaks. However, there may be a crystallization peak around 1190 °C, but it appears as not sharp with the increasing of the heating rate. While, the MBG has only one crystallization peak. In addition, as shown in Table 4, with the heating rate increasing, the crystallization peak temperature *T*_p_ also increases gradually. This should be related to the thermal hysteresis effect at a higher heating rate, which increases the crystallization temperature [18,19].

Glass hassome characteristic temperatures, such as glass transition temperature *T*_g_,crystallization initiation temperature *T*_x_, etc., which can be used to judge the thermal stability of glass. The more stable the glass is, the harder it is to crystallize. The difference ΔT between *T*_x_ and *T*_g_ can be used as the initial criterion of glass thermal stability. There is a weight parameter H’, which is H’ = (*T*_x_ − *T*_g_)*/T*_g_. It is known that the glass hasa better thermal stabilitywith the larger ΔT and H’ [20,21]. The obtained ΔT and H’ of the BG and MBG with the heating rate of 10 °C/min were given in Table 5. The higher the crystallization initiation temperature, the better the glass stability. As a result, the MBG has a better stability than the BG.

### 3.2. Crystallization Kinetics

In order to evaluate the thermal stability of the BG and MBG, the crystallization kinetics was calculated using the *T*_p_ obtained at different heating rates. According the JMA(Johnson-Mehl-Avrmi) equation [22,23,24,25]:(1)$$X_{t}=1-\exp\left[-{\left(kt\right)}^{n}\right]$$
where *X*_t_ is the volume fraction of glass phase transformed into a crystalline phase in the time of *t*, n is the growth index, *k* is the crystallization transition rate coefficient. The general expression equation of *k* is:(2)$$k=v\exp\left(-\frac{E}{RT}\right)$$
where *E* is the crystallization activation energy, *T* is the temperature, *v* is the frequency factor, and R is the gas constant.

Usually, Kissinger and Owaza are used to calculate the crystallization activation energy. In the Kissinger equation, the relationship between *T*_p_ and *α* is [26,27]:(3)$$\ln\frac{T_{p}^{2}}{\alpha }=\frac{E}{RT_{p}}+\ln\frac{E}{R}-\ln v$$
where *α* is the heating rate of DSC, *T*_p_ is the temperature of crystallization exothermic peak on the DSC curve, and *v* is the frequency factor. The activation energy of crystallization *E* under the Kissinger formula can be obtained.

The crystallization activation energy is the activation energy that overcomes the energy barrier during the structural units, rearranged as the glass melt transforms from a glass state to a crystalline state. The lower the barrier, the smaller the activation energy of the crystallization is, and the easier it is for glass to crystallize. The results obtained byfitting straight lines for different basalt glasses are shown in Table 6 and Figure 4. The higher the *T*_p_, the more difficult the crystallization. The activation energy of crystallization of the MBG is far greater than that of the BG, which means that the stability of the MBG is much better than that of BG.

Based on the obtained crystallization activation energy, the crystal growth index (n) can be well approximated by the Augis-Bennett Equation [28,29]:(4)$$n=\frac{2.5}{\Delta T}\times \frac{RT_{p}}{E}$$
where Δ*T* is the half maximum temperature of the DSC crystallization exothermic peak. According to the theory of solid phase transformation, glass has two crystallization modes: Surface crystallization and volume crystallization. The crystal growth index (n) can reflect the difficulty of glass crystallization and the glass crystallization mode. Generally, the larger the value of (n) is, the easier it is to crystallize and the more unstable the glass is. When 0 < n < 3, the crystallization mode of glass is surface crystallization, while when n ≥ 3, the crystallization mode of glass is volume crystallization [30,31,32].

The crystal growth index (n) of the BG and MBG is shown in Table 7. The crystal growth index (n) of the same basalt glass decreases with the increase of crystallization temperature, and it is less easy to crystallize. The crystal growth index (n) of the MBG is much smaller than that of the BG, which indicates that the MBG is more difficult to crystallize than BG and has a better thermalstability. The crystal growth index (n) of the BG and MBG is from 0.63 to 1.87, which is less than 3, so their crystallization mode is mainly surface crystallization.

### 3.3. XRD of Crystalline Phase

According to the crystallization temperature obtained by the DSC analysis curve, the BG and MBG were heated to 890, 1100, and 1190 °C at a heating rate of 10 °C/min, in addition,the MBG was heated to 1150 °C and kept for 3 h for the crystallization heat treatment. Figure 5 shows the XRD pattern of the BG after the crystallization heat treatment. At 1190 °C, the BG mainly precipitates iron magnesium oxide (Fe_2_O_3_) and a small amount of the feldspar phase. In addition, it mainly precipitates the iron phase and feldspar phase at 1100 °C. When it is at 890 °C, it mainly precipitates the pyroxene phase, a small amount of the feldspar phase and Fe_2_O_3_. Therefore, the general crystallization law of the BG can be obtained by Fe_2_O_3_ and (Fe,Mg)_2_O_3_ at a high temperature preferentially. Meanwhile, iron oxides are used as nucleating agents in the production of basalt cast stones and glass-ceramics [6,33,34,35,36]. Iron oxide is used as a nucleating agent because the bond energy of the Fe-O bond is smaller than that of the Si-O bond, and the bonding between the iron ions and oxygen destroys the tetrahedral network structure of silicon and oxygen, which makes nucleation easier by forming iron-rich aggregation areas [37]. Fe_2_O_3_ as a nucleating agent promotes the precipitation of the feldspar phase. With the decrease of crystallization temperature, the pyroxene phase is precipitated finally. Therefore, it can be considered that the Fe_2_O_3_ content in basalt glass is one of the key factors affecting its crystallization. For the MBG sample (Figure 6), a small amount of the peak was observed, which can be assigned to augite (Ca(Mg,Fe,Al)((Si,Al)_2_O_6_)). The reason is that the content of TFe in BG is reduced from 14.01% to 7.62% by modification, thus inhibiting the crystallization of the MBG. Therefore, modification is an effective way to devitrify basalt glass.

### 3.4. SEM of the Crystalline Phase

Basalt is a kind of rock with a dense or foam structure formed by the cooling and solidification of magma erupted by volcanoes. The magma cooling thermal history determines the degree of crystallization and the size of the basalt crystals. According to the crystallization degree of basalt, basalt structure can be divided into three types: Full crystalline structure, semi-crystalline structure, and vitreous structure. Additionally, basalt glass belongs to the vitreous structure, which is a thermodynamic unstable state and always tends to devitrify. Generally, the crystallization process will go through the process of nucleation → crystallite → skeleton → microcrystal. After nucleation, the crystal will grow according to its own structure and crystallization habit. However, due to the high viscosity of basalt glass melt and the slow diffusion and migration rate of substances, the substances needed for crystal growth are not fully supplied. Therefore, the whiskers of fast-growing crystals grow fastest on the vertical crystal plane, followed by crystal edges and corners. If crystal edges, corners, and crystal planes parallel to the growth direction are developed first, microcrystals with a relatively complete morphology and well-developed edges and planes can be formed, or skeleton crystals with a relatively depressed center of crystal planes can be formed [38,39].

Figure 7 shows SEM images of the heat-treated BG. The composition of thecrystallized phases(①−⑬) wereanalyzed using EDX and the results weregiven in Table 8. At 890 °C (Figure 7a), some granular skeleton crystals are precipitated from the BG. According to the atomic composition of crystalline phase ① and XRD, the crystallized phase should be pyroxene. At 1100 °C (Figure 7b), there are mainly five kinds of crystallized phases. According to the atomic composition of crystalline phase ① and XRD, crystallized phase ② is the skeleton of annular magnetite (Fe_3_O_4_), crystallized phase ③ is the skeleton of annular iron phase (Fe_2_O_3_) with ilmenite FeTiO_3_ coexisting, crystallized phase ④ is the chain feldspar cluster crystallite, crystallized phase ⑤ is the granular feldspar crystallite, and crystallized phase ⑥ is acicular sillimanite. At 1190 °C (Figure 7c), due to the high crystallization temperature, the viscosity of the basalt glass melt is relatively small, and the crystalline phase has skeletal crystals and crystallites with a relatively well developed crystal edge and crystal face. In the analysis, according to the atomic composition of crystalline phase ① and XRD, crystalline phase ⑦ is the iron phase (Fe_2_O_3_) microcrystal, crystalline phases (⑧, ⑨, and ⑩) are the magnesium aluminum iron plate microcrystals, which are the isomorphic substitution of iron phase Fe_2_O_3_, and crystalline phases (⑪, ⑫, and ⑬) are the magnesium oxide iron skeleton crystals, which are the isomorphic substitution of iron phase Fe_2_O_3_.

Figure 8 shows the SEM images of MBG after the crystallization heat treatment, the energy spectrum analysis of atomic composition results of crystalline phase⑭ and ⑮were given in Table 9. Compared withBG, the crystallization ability of MBG is greatly reduced, so that it is difficult to detect the crystalline phase by XRD, but a small amount of crystalline phase can still be observed with SEM. At 890 °C, there is only a small amount of crystallization. Crystalline phase ⑭, whose atomic composition results fromthe reference energy spectrum analysis, can be considered as poorly developed granular pyroxene crystals. At 1100 °C, a small amount of spherical grains precipitated, and the grains grew further at 1150 °C. However, almost no crystalline phase precipitated at 1190 °C. In comparison, it can be seen from Figure 8 that the crystallization ability of MBG after the heat treatment at 1150 °C is the strongest, which is consistent with the result of DSC. The crystallized phase ⑮ at 1150 °C should be a globular feldspar crystallite, according to the atomic composition of the crystallized phase in Table 8.

### 3.5. Discussion

In iron-rich silicate glasses, Fe_2_O_3_ can change the glass network and reduce the strength of glass connection as a result of chemical reactions between the isolated Si–O tetrahedron and Fe^3+^ion. Weakening of the glass network may facilitate atom diffusion, promoting nucleation and growth of the diopside crystal. The initial melts that are characterized by a high liquid phase are richer and promote the formation of the spinels as the initial crystal phase. By the prolonged thermal treatment, the pyrozene solid solution precipitates on the spinel (magnetite) crystals, which act as nuclei for the crystallization. In this work, the chemical composition of basalt glass was modified by adding a proper amount of pyrophyllite, diopside, and calcium carbonate. The content of iron has been reduced by nearly half. It is known that the Fe_2_O_3_ content had a strong depolymerization influence on the glass network. The reduction of the iron will improve the glass structure and enhance the crystallization activation energy. In addition, the number of formed nuclei will reduce. All of these effects will suppress the crystallization tendency and enhance the thermal stability of modified basalt glass.

## 4. Conclusions

The MBG was obtained by reducing the TFe content of BG. The crystallization activation energies ofBG and MBG were calculated by the Kissinger method. For BG, the activation energy of crystallization at 890 °C is 251.91 kJ/mol and at 1100 °C is 699.14 kJ/mol, while the activation energy of crystallization of the MBG at 1150 °C is 1432.90 kJ/mol. The activation energy of crystallization increases with the increase of temperature, and the activation energy of crystallization of the MBG is far larger than that of BG. According to the crystallization activation energy obtained by the Kissinger method, the average crystal growth index of BG is 1.87 and 1.41, respectively, and the average crystal growth index of MBG is 0.63. All of the average crystal growth indexes are less than 3, so their crystallization mode is mainly surface crystallization. 

The MBG has a better thermalstability than BG, which indicates that reducing the TFe content of basalt glass can effectively suppress its crystallization and improve the thermal stability of basalt glass. It provides an effective way for the large-scale and stable production of basalt fiber in the industry.

## Figures and Tables

**Figure 1 materials-13-05043-f001:**
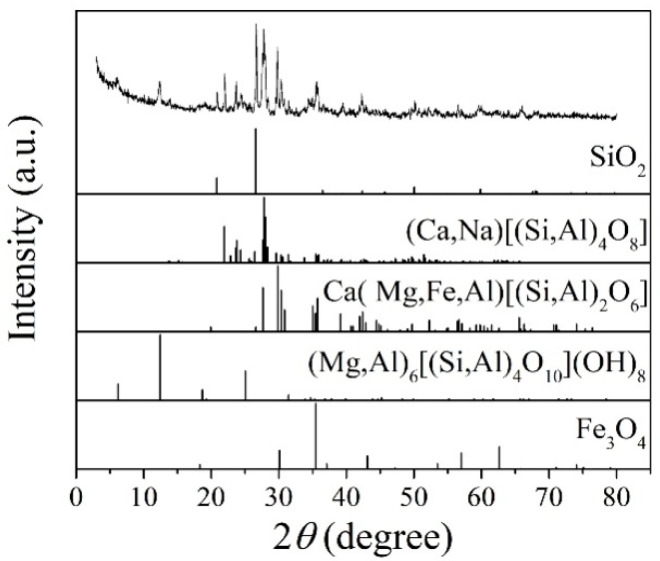
XRD patterns of the basalt rocks.

**Figure 2 materials-13-05043-f002:**
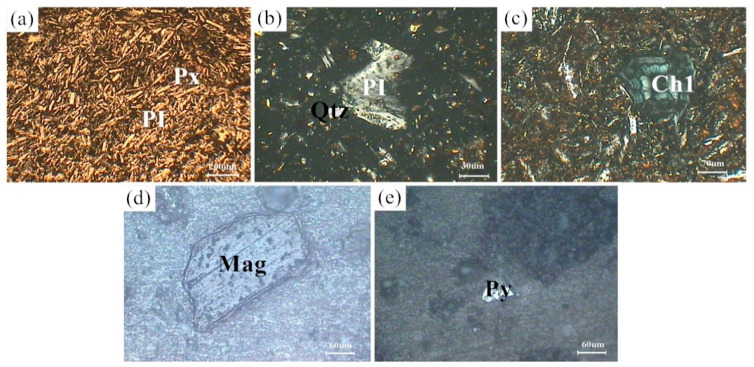
Microscopic features of the basalt rocks. (**a**) Px and PI; (**b**) PI and Qtz; (**c**) Ch1; (**d**) Mag; (**e**) Py.

**Figure 3 materials-13-05043-f003:**
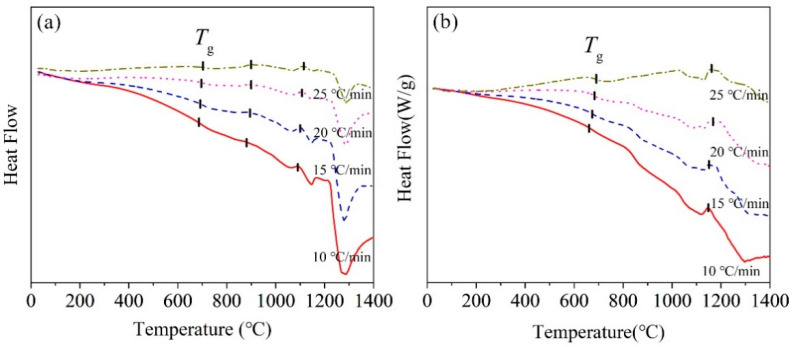
DSC curves of the (**a**) BG and (**b**) MBG measuredat different heating rates.

**Figure 4 materials-13-05043-f004:**
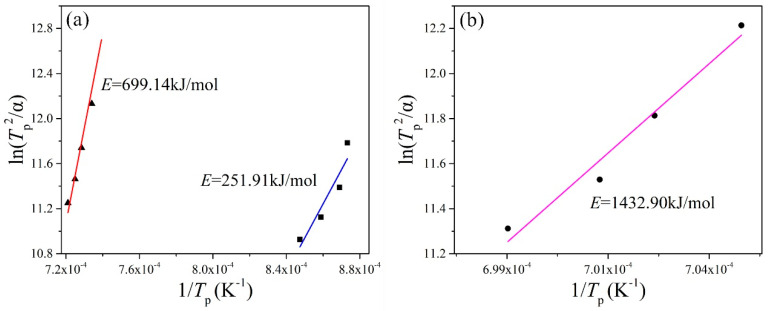
ln(*T*^2^_p_/*α*)~1/*T*_p_ fitting line of the (**a**) BG and (**b**) MBG.

**Figure 5 materials-13-05043-f005:**
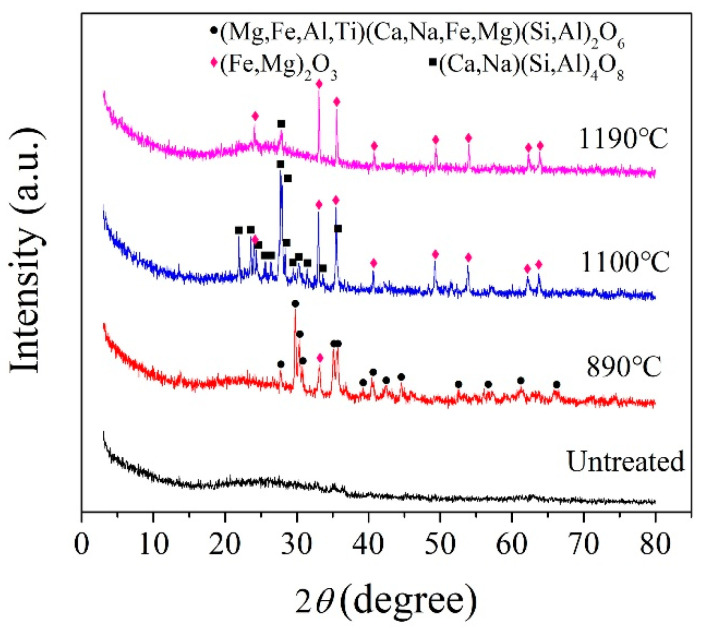
XRD patterns of the BG obtained at different crystallization temperatures.

**Figure 6 materials-13-05043-f006:**
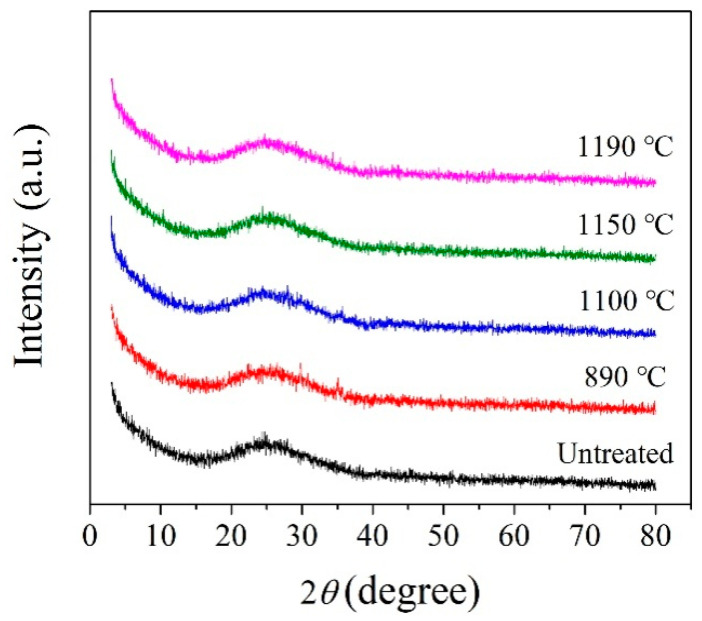
XRD patterns of the MBG obtained at different crystallization temperatures.

**Figure 7 materials-13-05043-f007:**
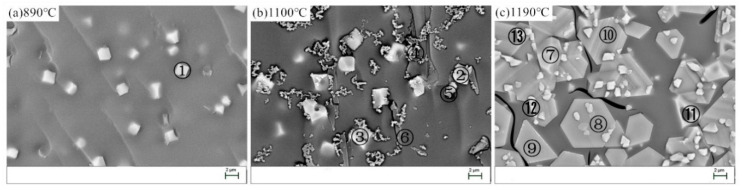
SEM images of the BG after the heat-treatment. (**a**) 890 °C; (**b**) 1100 °C; (**c**) 1190° C.

**Figure 8 materials-13-05043-f008:**
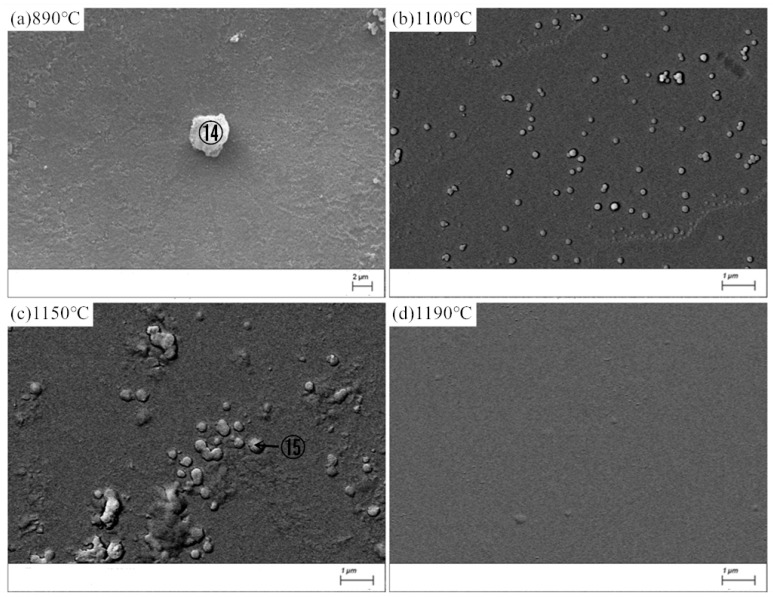
SEM images of the MBG after the heat-treatment. (**a**) 890 °C; (**b**) 1100 °C; (**c**) 1150 °C; (**d**) 1190 °C.

**Table 1 materials-13-05043-t001:** Chemical composition of the raw materials.

Raw Materials	Chemical Components, wt%							
	SiO_2_	Al_2_O_3_	TFe	CaO	MgO	Na_2_O + K_2_O	TiO_2_	Others
Basalt	49.87	15.11	13.61	9.34	3.91	3.46	1.97	2.73
Pyrophyllite	68.58	22.82	-	-	-	0.35	-	8.25
Diopside	59.15	1.96	0.25	21.50	13.10	0.40	0.89	2.75
Calcium carbonate	-	-	-	55.88	-	-	-	-

**Table 2 materials-13-05043-t002:** Composition of the basalt (B), basalt glass (BG), and modified basalt glass (MBG).

Samples	Chemical Composition, wt%							
	SiO_2_	Al_2_O_3_	TFe	CaO	MgO	Na_2_O	K_2_O	TiO_2_
B	49.87	15.11	13.61	9.34	3.91	2.40	1.06	1.97
BG	50.10	15. 28	14.01	9.44	4.09	2.51	1.21	2.12
MBG-1	54.32	14.27	9.62	12.17	4.25	1.54	0.98	1.62
MBG-2	54.90	14.50	7.62	12.60	4.80	1.25	0.96	1.83
MBG-3	54.65	15.83	5.72	12.42	4.98	1.35	1.01	1.93

**Table 3 materials-13-05043-t003:** Chemical composition of the MBG-2.

Samples	Si_2_O	Al_2_O_3_	TFe	CaO	MgO	Na_2_O + K_2_O	TiO_2_
Basalt (55%)	27.43	8.31	7.48	5.14	2.15	1.90	1.08
Pyrophyllite (25%)	17.15	5.71	-	-	-	0.09	-
Diopside (17%)	10.06	0.33	0.04	3.66	2.23	0.07	0.15
Calcium carbonate (3%)	-	-	-	2.99	-	-	-
MBG	54.64	14.35	7.52	11.79	4.38	2.06	1.23

**Table 4 materials-13-05043-t004:** The obtained crystallization temperature (*T*_p_) of the BG and MBG.

Samples	*T*_p_/°C			
	10 °C/min	15 °C/min	20 °C/min	25 °C/min
BG	872	877.68	891.26	906.91
	1089.09	1099.57	1106.13	1113.58
MBG	1146.23	1150.22	1152.75	1157.01

**Table 5 materials-13-05043-t005:** Characteristic temperature and stability parameter of the BG and MBG measured at 10 °C/min.

Sample	*T*_x_/°C	*T*_g_/°C	ΔT/°C	H’
BG	843.7	698.54	145.16	0.15
	1059.5	-	360.96	0.37
MBG	1118.88	620.67	498.21	0.56

**Table 6 materials-13-05043-t006:** Crystallization activation energy (*E*) of the BG and MBG at different temperatures.

Sample	*E* (kJ/mol)	
BG	251.91	699.14
MBG	1432.90	-

**Table 7 materials-13-05043-t007:** Crystal growth index (n) of the BG and MBG.

Sample	n				Average
	10 °C/min	15 °C/min	20 °C/min	25 °C/min	
BG	1.43	1.86	1.91	2.26	1.87
	1.28	1.35	1.44	1.55	1.41
MBG	1.01	0.62	0.44	0.44	0.63

**Table 8 materials-13-05043-t008:** Chemical compositions of the crystalline phase in the BG.

Element	Atomic%												
	①	②	③	④	⑤	⑥	⑦	⑧	⑨	⑩	⑪	⑫	⑬
O	30.96	57.21	60.02	40.07	48.44	60.94	62.36	57.64	54.36	59.68	53.17	56.24	52.43
Mg	1.91	/	/	/	/	/	/	12.09	12.72	10.17	15.48	16.95	14.70
Al	/	/	/	13.99	9.51	9.86	/	4.08	4.65	3.96	/	/	/
Si	17.19	/		31.13	31.07	29.21	/	/	/	/	/	/	/
Ca	49.94	/	/	14.81	10.98	/	/	/	/	/	/	/	/
Ti	/	/	6.06	/	/	/	/	/	/	/	/	/	/
Fe	/	42.79	33.92	/	/	/	37.64	26.19	28.28	26.18	31.35	26.80	32.84

**Table 9 materials-13-05043-t009:** Chemical compositions of the crystalline phase in the MBG.

Element	Atomic%	
	⑭	⑮
O	64.25	53.70
Mg	2.29	/
Al	4.13	5.74
Si	16.33	32.51
Ca	12.99	8.05

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
