# Peer review of "Crystallization Kinetics of Modified Basalt Glass"

_materials, 2020, doi:10.3390/ma13215043_

Round 1

Reviewer 1 Report

In this manuscript, the authors analyse the characteristics of the crystallization process of a basalt glass and a “modified” basalt glass, concluding that the modified counterpart exhibits inhibited crystallization and that crystallization proceeds from the surface.

Unfortunately, the manuscript presents too strong flaws to be accepted for publication in “Materials”. The main one is the lack of discussion. The authors show some experimental results, as obtained from calorimetry, x-ray and electronic microscopy to claim that the modified basalt glass crystalizes more slowly than the normal basalt glass, but the discussion ends there. They do not discuss or explain why is this behaviour observed nor what are the underlying physical causes.

Apart from this main concern, other issues, equally important, affects the quality of the research work. For example, from the experimental point of view, the research is quite poor, since, as far as I see, they only show results on one basalt glass and one modified basalt glass, without explanation of the specific characteristics of the “modified” basalt glass, or without explanation of the experimental protocol to produce it. At some point, they explain that “at the same time, by increasing the content of Fe2O3, the MBG has better stability than the BG”. However, I do not see results on the stability of the BMG as a function of Fe2O3 content. Also, the authors make claims that are, in my opinion, not well supported, for example

- lines 96-98, when they mention the mineral composition of the basalt. Some further analysis or reference would be required.

- line 108, “not sharp because of the increase of the heating rate”. Should not we expect the opposite trend? If the rate is faster, the release of enthalpy is more abrupt, so the peak would be sharper and narrower….

- line 109, “the shape of the crystallization peaks are flat and wide, which indicates that the surface crystallization of fine crystal grains” (sic). Some reference is required….

- line 118, “the more stable the glass is, the easier is to crystallize”. I assume this is a mistake, because later they contradict themselves: “the MBG has better stability than the BG, which means the MBG is less likely to be crystallized”.

- The use an expression that contains the width of the peaks. However, peaks, if any, are very broad and diffuse. I would like to know how the authors obtain the value of the peak’s width.

- I do not really understand what is the relevance of the description of the crystals they find in each material in relation with the conclusion of the paper. Further explanation on that would be required.

Finally, and equally important as well, the writing style of the manuscript (and grammar) is extremely poor. It gives the impression that the manuscript has been written in a rush. Some sentences that made me reach this conclusion are:

  • “[…] it appears not sharp because of the increase of the heating And the MBG has only one crystallization peak.”
  • “[…] are flat and wide, which indicates that the surface crystallization of fine crystal grains.”
  • “On the basis of ΔT, there is a weight parameters H' , which H' =(Tx -Tg) /Tg.”
  • “Crystallization activation energy is when glass melt transforms from glass state to the crystalline state, a certain activation energy is required to overcome the energy barrier when the structural units are rearranged.”
  • “XRD patterns of the (a) BG and (b) MBG obtained at different crystallization temperatures for crystallizatio n temperature.”

Apart from grammar mistakes, lack of punctuation signs and missing information in figure captions.

Overall, I find this manuscript is of very low quality, and it must be improved in terms of discussion, experimental design and format to be recommended for publication.

Reviewer 2 Report

The manuscript requires revision.

My questions and suggestions are listed below.

Line 22. Please specify that you mean the MBG when you write that “a small amount of the crystalline phase can be observed”.

Line 29. There is no keyword “Crystallization phase”. Please use the words “crystalline phase”. Please make correction throughout the manuscript.

Line 51. Please explain the symbol “TFe2O3”.

Lines 54, 55. Please explain what spinel, peridot and iron and titanium oxide you mean or add their chemical formulas.

Lines 55, 57, 60. Please make references in the following manner: “Walker et al. [7]”.

Line 57. Please be aware that Olivier is a first name of Olivier Namur, his surname is Namur. Please make correction.

Line 62. Please use subscript in the Sio2 formula.

Line 76. Please explain where you got Basalt Glass and Modified Basalt Glass or how you prepared them.

Line 84. Did you use the powdered or bulk samples? What was the weight of the sample?

Line 95. Please start the word “lithofacies” with the capital letter L.

Lines 95 – 104. Please explain the reasons for the study of the lithofacies and phase composition of the basalt. How these results are connected with the study of the Basalt Glass and Modified Basalt Glass crystallization kinetics?

Line 102. Please provide the chemical formulas of plagioclase, augite, quartz, chlorite and magnetite and provide the numbers of the reference XRD cards.

Line 114. Fig. 3. Please indicate Tg in all the DSC curves of your glasses.

Line 115. Table 2. Please explain the meaning of the letters L, M, and H.

Line 117. “The more stable the glass is, the easier it is to crystallize.” I think it is incorrect.

Line 135. Please explain the meaning of the words “thermodynamic temperature”.

Line 175. Please correct the phrase “magnesium oxide iron”. There is no such composition.

Line 178. Please provide the formula of the “iron phase”.

Line 179. Please provide the formula of the “Fe2O3 isomorphic substituted magnesium iron oxide”.

Line 187. I cannot understand the meaning of the phrase “the XRD diffraction peaks are all in a dispersed state”.

Line 195. “If the content of the crystalline phase is less than 5%, the existence of the crystalline phase cannot be detected by XRD.” It is incorrect as a general conclusion. Each element has a different atomic scattering factor, which represents how strongly x-rays interact with those atoms. So the detection limit is different for different compounds.

Fig. 5. Please provide the numbers of the reference XRD cards.

Line 205. Generally, the crystallization process will go through the process of nucleation → crystallite → skeleton → microcrystal. Do you think skeleton is always formed during the crystallization process?

There is a great discrepancy between the SEM images and XRD patterns of your materials. Please explain how you prepare samples for the XRD study. Do you use powders or make the XRD analysis of the crystallized surface of your materials?

References.

Please prepare all the references in the same style.

Line 279. Olivier; Bernard; and Vander are not the surnames. Please correct the reference.  

Line 282. Harold is not the surnames. Please correct the reference.

The grammar correction is required throughout the manuscript.

Please correct sentences or words written on lines 22-24;line 174, line 214.

Reviewer 3 Report

I have attached my comments or observations from the review of the article “Crystallization Kinetics of Modified Basalts Glass”. the aim of the authors is to analyse the kinetics of crystallization and to analyse the compositional factors that may ease it

In my opinion, the work is interesting and can give the community a better understanding of the production process of basalts glass. The authors have carried out a good work based on the control of the crystallization behaviour in order to stabilize the structure of basalts glass and, furthermore on the analysis of their characteristic kinetic parameters. The authors, also, have done a thorough study on the structural, thermal and morphologic characterization of basalts with different composition to achieve its objective.

However, I suggest for the publication of the article some modifications:

1.- In general, the figures are too compacted. Connecting the text to a figure is complicated.

2.-The explanation in Fig. I and Fig. 2 are too schematic and is not understood. I would suggest modifying the wording that explains these figures in its entirety in the text and, also, in the figure captions.

3.- I also suggest reading the whole article to correct any typographical errors in the text

My conclusion is that this article could be published if the proposed modification is carried out previously.

Round 2

Reviewer 1 Report

After the inclusion of plenty of new information, I think the manuscript can be accepted for publication. However, before that, the text should be reviewed to check for the various mistakes and incomplete sentences as a result of the incorporation of the new information.

Reviewer 2 Report

Careful proofreading is required. In many cases, addition of new sentences causes appearance of meaningless text. Just as an example, line 11, “The larger the ΔT and H', the better the of The glass 11 would has good thermal stability with larger ΔT and H'[20,21]”, and in figure captio, line 87, “XRD patterns of the BG obtained at different crystallization temperatures for crystallization 87 temperature.”

In Table 1, the sum of the components of the individual raw materials do not give total 100%.

Do you mean that the content of CaO in calcium carbonate is 99.80 wt% (Table 1)?

In Table 1, you use wt%, while in Table 2 - mass fraction, %. Please explain the difference.

  1. 3. “About 50 g of the mixture was heated in alumina crucibles to dissociate the carbonates and borate...” Please explain how borates appeared in the mixture.

In Table 7, instead of designation L, M, and H please provide the values of the heating rate.

Please provide numbers for figures with XRD patterns.

References

Line 191. Vander is not a family name.
